# A Simple Way to Learn Metrics Between Attributed Graphs

**Yacouba Kaloga**
Univ Lyon, Ens de Lyon, CNRS,
Laboratoire de Physique, UMR 5672
Lyon, France
`yacouba.kaloga@ens-lyon.fr`

**Pierre Borgnat**
Univ Lyon, Ens de Lyon, CNRS,
Laboratoire de Physique, UMR 5672
Lyon, France
`pierre.borgnat@ens-lyon.fr`

**Amaury Habrard**
University of Lyon, UJM-Saint-Etienne,
CNRS, Laboratoire Hubert Curien, UMR 5516,
Saint-Etienne, France
Institut Universitaire de France (IUF)
`amaury.habrard@univ-st-etienne.fr`

## Abstract

The choice of good distances and similarity measures between objects is important for many machine learning methods. Therefore, many metric learning algorithms have been developed in recent years, mainly for Euclidean data, in order to improve performance of classification or clustering methods. However, due to difficulties in establishing computable, efficient and differentiable distances between attributed graphs, few metric learning algorithms adapted to graphs have been developed despite the strong interest of the community. In this paper, we address this issue by proposing a new Simple Graph Metric Learning - SGML - model with few trainable parameters based on Simple Graph Convolutional Neural Networks - SGCN - and elements of Optimal Transport theory. This model allows us to build an appropriate distance from a database of labeled (attributed) graphs to improve the performance of simple classification algorithms such as $k$-NN. This distance can be quickly trained while maintaining good performance as illustrated by the experimental studies presented in this paper.

## 1 Introduction

Classification of attributed graphs has received much attention in recent years because graphs are well suited to represent a broad class of data in fields such as chemistry, biology, computer science, etc [1, 2]. Advances were obtained, particularly thanks to the development of graph convolutional neural networks (GCN) [3–6] of which many actually graph learning model can rely on [7, 8]. GCN has attracted interest in the recent years, due to their low computational cost, their ability to extract task-specific information, and their ease of training and integration into various models. Some tackle classification problems for attributed graphs by leveraging GCN: they characterize and build Euclidean representations for attributed graphs both in a supervised (e.g. [5, 9]) or unsupervised (e.g. [10, 11]) way. Despite these achievements, classification methods based on direct evaluation of similarity measures between graphs remain relevant since they can obtain similar, and in some cases even better, performance [12]. Currently, most of these methods work in a task-agnostic way. However, given the diversity of graph datasets, we can not expect one similarity measure to be well suited for all of them, on all learning tasks.

Adapting similarity measures to specific datasets and related tasks help to improve their generality and their performance. One of such approach is known as Metric Learning (hereafter ML), and has already been successful for Euclidean data. Xing et al. [13] were the first to propose a Metric Learning method to improve a specific method and task ($k$-means for clustering of Euclidean data). This first

Y. Kaloga et al., A Simple Way to Learn Metrics Between Attributed Graphs. *Proceedings of the First Learning on Graphs Conference (LoG 2022)*, PMLR 198, Virtual Event, December 9–12, 2022.

work sparked a strong interest in ML which led to the development of many methods [14–17] for Euclidean data. In contrast, few of these methods exist for attributed graphs. Existing methods (e.g., [18]) rely on iterative procedures which are hardly differentiable, and this makes also scalability an issue. In the state-of-the-art of classification, neural networks tend to currently dominate in the literature, yet building simple and learned (hence adapted to data and task) similarity measures between attributed graphs remain a relevant issue for at least two reasons: it allows to step up simpler graph classification algorithms, and also it allows to rely on graph kernels [1, 19] which are, as of today, as efficient on numerous tasks as models relying on graph neural networks.

**Our contribution.** To address the issue of scalability in Metric Learning for graphs, we propose here a novel graph ML method, called Simple Graph Metric Learning (SGML). In the first step, attributed graphs are coded as distributions by combining the attributes and the topology thanks to GCN. Then, relying on Optimal Transport, we define a novel metric between these distributions, that we call Restricted Projected Wasserstein, $\mathcal{RPW}_2$ for short. $\mathcal{RPW}_2$ is differentiable and has a quasi-linear complexity on the distribution size (in number of bins; this is also the number of nodes); it removes certain limitations of the well known Sliced Wasserstein (noted $\mathcal{SW}_2$) [20]. $\mathcal{RPW}_2$ is then used to build a parametric pseudo-metric between attributed graphs which then has also a quasi-linear complexity on the graph size (in the number of nodes). The similarity measure proposed in SGML has a limited number of parameters, and it helps the model to scale efficiently. Next, we focus on the the k-nearest neighbors (k-NN) method for classification. An advantage of using k-NN is that, if the learning set grows, one can exploit it at near zero additional cost (since it only requires to store these new data) on the contrary of SVMs that would require to retrain the whole data (a task quadratic in size). Since many real datasets (e.g., graphs from social networks, or to detect anomalies on computer networks) are expected to have a growing size, this property is important for continual learning, and from an energetic and environmental stance to avoid costly retraining. In order to use k-NN and train the distance, we propose a novel softmax-based loss function over class point clouds. It appears to be novel in the context of graph ML and it leads to better results in the explored setting than the usual ML losses (i.e., those specifically built to improve k-NN for Euclidean data). Our experiments show that SGML learns a metric increasing significantly the k-NN performance, compared to state-of-art algorithms for graph similarity measures.

The article is organized as follows. In Section 2, we discuss related works on graph metric learning and on optimal transport theory applied to the construction of attributed graphs similarity measures. Section 3 provides useful notations and definitions needed for the present work. The SGML model is defined in Section 4. Finally, in Section 5, we present various numerical experiments assessing the efficiency of our model. These experiments show that in various conditions, SGML has great ability to build accurate distance with competitive performance with the state-of-the-art in classification of graphs, both in context of k-NN and kernel-based methods, and that despite its limited number of parameters. A main advantage of the proposed SGML method is also its simplicity, hence leading to a scalable and efficient method for graph Metric Learning. We conclude in Section 6.
**Societal Impact** The contribution is essentially fundamental, and we do not see any direct and immediate potential negative societal impact. Conversely, the scalability of the method will help to alleviate the energy consumption of ML on graphs.

## 2 Related Works

### 2.1 Graph Metric Learning

About ML for graphs, we can notably mention a series of works [21–23] that consist in learning a metric through Graph Edit Distance (GED). The major disadvantage of these methods is the complexity of the computation of the GED which can be only done for very small graphs.

Following the introduction of GCN, an approach based on Siamese neural networks has been proposed in [24] for the study of brain connectivity signals, represented as graphs signals. In this specific case, all graphs are the same and they differ only by the signal they carry. This makes this method not applicable to most of datasets. More recently models without neural networks have been proposed: [18] present *Interpretable Graph Metric Learning* which builds a similarity measure by counting the most relevant subgraphs to perform a classification task. However, their method cannot handle large graphs. [25] proposes to learn a kernel based on graph persistent homology. The resulting model is also efficient, but it has the disadvantage of not being able to deal with discrete features in graphs.

As seen, existing work on graph ML are either limited by the assumptions made to build their model, or too costly, or not suitable to actually leverage simple (classification) algorithms and increase their performance. To obtain a simple graph ML procedure that is not itself too costly, we need to have a similarity measure between graphs that can be computed quickly. To construct such a distance, recent works suggest that Optimal Transport is an appropriate tool.

### 2.2 Optimal Transport for Graphs

Optimal Transport (OT) has been put forward as a good approach to quickly compute similarity measures between graphs, relying on the the fact that it provides tools for computing metric between distributions [26]. Recent studies have shown that efficient distances and kernels for graphs can be built from this theory. Fused-Gromov-Wasserstein [27] is such a metric (distance in a mathematical sense) using OT to compare graphs through both their structures and attributes. Notably it allows one to compute barycenter of a set of graphs, and interpolation between graphs. Experimentally, it leads to good results in classification. Its bi-quadratic complexity in the size of graphs is its main drawback, even if it can be reduced to cubic cost with entropic regularization.

In [28] an OT based approach to compare graphs is developed. It uses OT between specific signals on the graphs. Thanks to a Gaussian distribution hypothesis, the analytical expression of the OT between these signals is derived. While the model provides good results, it is initially limited to graphs having the same size, and a task of node alignment (which has a cubic complexity) must be performed. [29] relaxes the condition on size, yet the focus remains on graph alignment of non attributed graphs.

[12] has proposed the Wasserstein Weisfeiler-Lehman (WWL) method which can be seen as an evolution of the previous work [28] without the two hypotheses, neither on the size of the graphs nor on the specificity of the graph signals. In addition, a non trainable GCN is used to build task-agnostic characteristics which are then compared through OT. This pseudo-metric is then used to build an efficient kernel for graph classification. Unfortunately this model requires the computation of the optimal transport map which has a cubic cost (or quadratic with entropy regularization).

While these models are efficient on classification tasks, their complexity remains high, and they are not fast enough (being quadratic or more) to be incorporated in a framework of Metric Learning. A part of our contribution is to provide such an optimal transport-based fast similarity measure for attributed graphs, with no restriction on the nature of the graphs (and their attributes) to be compared.

## 3 Background on Metric Learning and Optimal Transport

**Notations.** Let us consider a finite dataset $\mathbb{X} = \{\boldsymbol{x}_i\}_{i=1}^{|\mathbb{X}|}$ whose elements are in $\mathbb{R}^q$. The dataset comes with a set of labels $\mathbb{E} = \{e_i\}_{i=1}^{|\mathbb{E}|}$ and a labeling function $\mathcal{E} : \mathbb{X} \to \mathbb{E}$. We note $\mathcal{P}(\mathbb{X}) \subset \mathcal{P}(\mathbb{R}^q)$ the set of discrete probability over $\mathbb{X} \subset \mathbb{R}^q$. $\delta_{\boldsymbol{x}}$ is the Dirac distribution centered in $\boldsymbol{x}$. We note $d$ a metric on $\mathbb{X}$. It verifies the following properties: **Symmetry** - $\forall(\boldsymbol{x}, \boldsymbol{y}) \in \mathbb{X}^2, d(\boldsymbol{x}, \boldsymbol{y}) = d(\boldsymbol{y}, \boldsymbol{x})$; **Identity of indiscernibles** - $\forall(\boldsymbol{x}, \boldsymbol{y}) \in \mathbb{X}^2, d(\boldsymbol{x}, \boldsymbol{y}) = 0 \Leftrightarrow \boldsymbol{x} = \boldsymbol{y}$; **Triangle inequality** - $\forall(\boldsymbol{x}, \boldsymbol{y}, \boldsymbol{z}) \in \mathbb{X}^3, d(\boldsymbol{x}, \boldsymbol{z}) \leq d(\boldsymbol{x}, \boldsymbol{y}) + d(\boldsymbol{y}, \boldsymbol{z})$. $d$ is referred to as a pseudo-metric when it follows these properties except the identity of indiscernibles. In this article, the term "distance" will be used sometimes in an informal way as a synonym of discrepancy or measures of similarity. Additionally the term "distribution" will always refer to discrete distribution.

### 3.1 Learning a metric

For ML, we suppose that a dataset $\mathbb{X}$ is given with the knowledge of two sets: $\mathcal{S}$ (similar) and $\mathcal{D}$ (dissimilar), containing pairs of some elements of $\mathbb{X}$. The goal is to build a parametric distance $d_\theta$ in such a way that the pairs of elements in $\mathcal{S}$ should be *close* while the pairs in $\mathcal{D}$ should be *far away*[1]. These sets are often built from the labeling function of $\mathbb{X}$ such that $\{\boldsymbol{x}_i, \boldsymbol{x}_j\} \in \mathcal{S}$ if $\mathcal{E}(\boldsymbol{x}_i) = \mathcal{E}(\boldsymbol{x}_j)$ otherwise $\{\boldsymbol{x}_i, \boldsymbol{x}_j\} \in \mathcal{D}$. An optimization problem depending on $d_\theta$, $\mathcal{S}$ and $\mathcal{D}$ is then defined with a loss function $\mathcal{F}$ suitable for the purpose:

$$\max_\theta \mathcal{F}(d_\theta, \mathcal{S}, \mathcal{D}) \tag{1}$$

---

[1]Some algorithms use a third type of information, which consists of triples indicating that a given element must be closer to such element than to another element [16].

We denote $\theta^*$ the optimal parameters. The interest for building such a distance $d_{\theta^*}$ with respect to information in $\mathcal{D}$ and $\mathcal{S}$ lies in the fact that $\mathbb{X}$ is often included in a larger set, containing elements which are not labeled. The goal is that the obtained distance $d_{\theta^*}$ will ease learning algorithm to find these missing labels. A part of our work will be to introduce a new and suitable loss function $\mathcal{F}$ in metric learning literature for the problem of metric learning for graphs.

## 3.2 Optimal transport

Let us consider two finite datasets $\mathbb{X}$, $\mathbb{X}'$, and two distributions $\mu \in \mathcal{P}(\mathbb{X})$ et $\nu \in \mathcal{P}(\mathbb{X}')$ on these sets:

$$\mu = \sum_{\boldsymbol{x}_i \in \mathbb{X}} a_i \delta_{\boldsymbol{x}_i} \quad \text{and} \quad \nu = \sum_{\boldsymbol{x}'_i \in \mathbb{X}'} b_i \delta_{\boldsymbol{x}'_i} \tag{2}$$

with $a_i \geq 0$, $b_i \geq 0$, $n = |\mathbb{X}|$, $n' = |\mathbb{X}'|$, and $\sum_{i=1}^{n} a_i = 1$, $\sum_{i=1}^{n'} b_i = 1$. Given a continuous cost function $c : \mathbb{R}^q \times \mathbb{R}^q \to \mathbb{R}_+$, one can build from optimal transport a metric between distributions with support in $\mathbb{R}^q$, the so-called 2-Wasserstein distance $\mathcal{W}_2$ :

$$\mathcal{W}_2(\mu, \nu) = \inf_{\pi_{i,j} \in \Pi_{a,b}} \Big( \sum_{i,j=1}^{n,n'} \pi_{i,j} c(\boldsymbol{x}_i, \boldsymbol{x}'_j)^2 \Big)^{\frac{1}{2}} \tag{3}$$

$\Pi_{a,b}$ is the set of joint distributions on $\mathbb{X} \times \mathbb{X}'$, $\pi = \sum_{i,j=1}^{n,n'} \pi_{i,j} \delta_{(\boldsymbol{x}_i, \boldsymbol{x}'_j)}$ whose marginals are the distributions $\mu = \sum_{\boldsymbol{x}'_i \in \mathbb{X}'} \pi(\cdot, \boldsymbol{x}'_i)$ and $\nu = \sum_{\boldsymbol{x}_i \in \mathbb{X}} \pi(\boldsymbol{x}_i, \cdot)$. We note $\pi^* \in \Pi_{a,b}$ the optimal distribution (or coupling, or map) giving the solution of this problem. The cost function $c$ is taken as 2-norm: $c(\boldsymbol{x}_i, \boldsymbol{x}'_j) = \|\boldsymbol{x}_i - \boldsymbol{x}'_j\|_2$, leading hence to the 2-Wasserstein distance. This defines an efficient way to compare distributions. One could use differentiable versions (w.r.t the parameters of a distribution) by considering the 1-Wasserstein [30] or the entropic regularization of $\mathcal{W}_2$ [26, 31]. Still, they are not suitable for metric learning because of the (initial) complexity (when $n = n'$) in $O(n^3 \log n)$, or $O(n^2 \log(n))$ with entropic regularization thanks to the Sinkhorn algorithm [26].

**Sliced Wasserstein distance ($\mathcal{SW}_2$).** In order to drastically reduce the cost for computing the OT, [20] has proposed a modified metric $\mathcal{SW}_2$ which consists to compare the measures $\mu$ and $\nu$ via their one dimensional projections. Let $\boldsymbol{\theta} \in \mathbb{S}^{q-1}$ be a vector of the unit sphere of $\mathbb{R}^q$. Distributions $\mu$ and $\nu$ projected along $\boldsymbol{\theta}$ are denoted $\mu_{\boldsymbol{\theta}} = \sum_{\boldsymbol{x}_i \in \mathbb{X}} a_i \delta_{\boldsymbol{x}_i \cdot \boldsymbol{\theta}}$ and $\nu_{\boldsymbol{\theta}} = \sum_{x'_i \in \mathbb{X}'} b_i \delta_{\boldsymbol{x}_i \cdot \boldsymbol{\theta}}$. $\mathcal{SW}_2$ is defined as follows:

$$\mathcal{SW}_2(\mu, \nu)^2 = \int_{\mathbb{S}^{q-1}} \mathcal{W}_2(\mu_{\boldsymbol{\theta}}, \nu_{\boldsymbol{\theta}})^2 d\boldsymbol{\theta} \tag{4}$$

The advantage of this formulation stems from the quasi-linearity in $n$ (or $n'$) of the computation cost of $\mathcal{W}_2$ distance between one dimensional distributions. The integral can be estimated via a Monte-Carlo sampling. The complexity is then (when $n' \leq n$) at most $O(M(n \log n))$ with $M$ the number of samples (uniformly) drawn from $\mathbb{S}^{q-1}$. However, [32] shows that $\mathcal{SW}_2$ is a biased downwards compared to $\mathcal{W}_2$, since the vector $\boldsymbol{\theta}$ for projection determines at the same time the OT plans and also the cost of transport; this leads to a less effective distance.

**Projected Wasserstein distance ($\mathcal{PW}_2$).** When $n = n'$, $\mathcal{PW}_2$ is introduced by [32] in answer to previous limitations. $\mathcal{PW}_2$ is computed similarly as $\mathcal{SW}_2$, but for each projection $\boldsymbol{\theta}$, the one dimensional optimal transport plan $\pi^{\boldsymbol{\theta},*}$ between $\mu_{\boldsymbol{\theta}}$ and $\nu_{\boldsymbol{\theta}}$ is used with the original distributions $\mu$ and $\nu$ so as to compute the transport cost:

$$\mathcal{PW}_2(\mu, \nu)^2 = \int_{\mathbb{S}^{q-1}} \sum_{i,j=1}^{n,n'} \pi_{i,j}^{\boldsymbol{\theta},*} \|\boldsymbol{x}_i - \boldsymbol{x}'_j\|_2^2 d\boldsymbol{\theta} \tag{5}$$

They show that this formulation defines a metric, has good properties and is more suitable for several learning tasks, e.g. generative tasks or reinforcement learning. Unfortunately their result holds only for uniform distributions of the same size. Our method rely on an extended version of this definition, involving distributions of different sizes and not necessarily uniform.

# 4 A Simple and Scalable Graph Metric Learning

Let us consider a dataset $\mathbb{G}$ of attributed graphs with labeling set $\mathbb{E}$ and labeling function $\mathcal{E}$. For a given graph $\mathcal{G} \in \mathbb{G}$ having $\boldsymbol{A}$ as adjacency matrix, we call $n$ the number of node of the graph. Each node $i$ of $\mathcal{G}$ carry features $\boldsymbol{X}(i,:) \in \mathbb{R}^q$; thus $\boldsymbol{X} \in \mathbb{R}^{n \times q}$ is the matrix of attributes of the graph.

## 4.1 From graph to distribution

Previous works using OT (pseudo-)metric have shown that comparing graphs through the signal they carry is a good way to compare them; we follow this path. The first step of our learning method consists in the generation of features jointly representative of the structure of each graph $\mathcal{G}$ and the attributes of their nodes $\boldsymbol{X}$. We use for this purpose Simple GCN [6], a streamlined version of GCN in which all the intermediate non-linearities have been removed. This choice is dictated by the need to strongly reduce the number of trainable parameters, and it accelerates the training without degrading its performance compared to other GCN. This Simple GCN creates features as:

$$\boldsymbol{Y} = \text{ReLU}(\widetilde{\boldsymbol{A}}^r \boldsymbol{X} \boldsymbol{\Theta}) \tag{6}$$

where $\boldsymbol{X} \in \mathbb{R}^{n \times q}$ are the initial attributes of the nodes, $\widetilde{\boldsymbol{A}} = \boldsymbol{A} + \boldsymbol{I}_n$ (where $\boldsymbol{I}_n$ is the identity matrix of $\mathbb{R}^n$) and $\boldsymbol{Y} \in \mathbb{R}^{n \times p}$ are the features computed by SGCN. The neighborhood exploration depth $r$ of this GCN is one of the hyperparameters of the method, along with the dimension $p$ of the extracted features $\boldsymbol{Y}$. The coefficients of the matrix $\boldsymbol{\Theta} \in \mathbb{R}^{q \times p}$ of this GCN are the (only) trainable weights of the method. We will always choose $p \leq q$, so the method has at most $q^2$ trainable parameters. From the extracted features $\boldsymbol{Y}$, we define a uniform distribution whose suport is the nodes' characteristics:

$$\mathcal{D}_{\boldsymbol{\Theta}}(\mathcal{G}, \mathbf{X}) = \sum_{i=1}^{n} \frac{1}{n} \delta_{\boldsymbol{Y}(i,:)} \tag{7}$$

This first step is similar to WWL [12], except that we consider a trainable GCN, $\boldsymbol{\Theta}$ being the trainable parameters. In eq. (7), both the structure $\mathcal{G}$ and the attributes $\mathbf{X}$ are accounted for. Next, we propose a novel way to evaluate the similarity between attributed graphs using these distributions.

## 4.2 From distributions to distance

The distances between graphs are computed as a distance between their representative distributions (Eq. (7)) with OT; specifically, we propose a novel one, called Restricted Projected Wasserstein (and noted $\mathcal{RPW}_2$) extending $\mathcal{PW}_2$ previously introduced in [32].

**Restricted Projected Sliced-Wasserstein.** In [32], $\mathcal{PW}_2$ is only defined for uniform distributions when $n = n'$. We extend this definition to cases $n \neq n'$, and we remove the constraint of uniformity. We show in Appendix A.1 that it still defines a metric (on discrete distribution space).

In order to compute this quantity, we could rely on Monte-Carlo sampling, and the complexity would be $O(Mpn \log(n))$. This can be prohibitive due to the term $pM$. In order to have an even more scalable model, we restrict the projections to be alongside the basis vectors $\{\boldsymbol{u}_k\}_{k=1}^{p}$ of $\mathbb{R}^p$ only. This choice stems from a spanning constraint that allows us to define a quantity (named Restricted $\mathcal{PW}_2$, or $\mathcal{RPW}_2$ for short) verifying the identity of indiscernibles without increasing significantly the computing time (see Appendix A.1). This guarantees that $\mathcal{RPW}_2$ is also a metric of discrete distribution space as $\mathcal{PW}_2$ (see also Appendix A.1). Therefore it can always distinguish distributions that are different but also (associated with the continuity) that, when two distributions are getting closer, then $\mathcal{RPW}_2$ tends towards 0; this is important in ML context. $\mathcal{RPW}_2$ is expressed as:

$$\mathcal{RPW}_2(\mu, \nu)^2 = \frac{1}{p} \sum_{k=1}^{p} \sum_{i,j=1}^{n,n'} \pi_{i,j}^{\boldsymbol{u}_k, *} \|\boldsymbol{x}_i - \boldsymbol{x}_j'\|_2^2 \tag{8}$$

Note that $\mathcal{RPW}_2$ defines a metric therefore it should considered as such and not as an attempt to approximate $\mathcal{PW}_2$. A major advantage of $\mathcal{RPW}_2$ is that it is defined by a deterministic formula; this avoids the variability introduced by a Monte-Carlo sampling (when one would need to evaluate $\mathcal{PW}_2$ or $\mathcal{SW}_2$). Anyway, we can notice that for a given $\boldsymbol{u}_k$, many $\pi_{i,j}^{\boldsymbol{u}_k}$ may be optimal for the projected distribution on $\boldsymbol{u}_k$, while they may lead to different values when computing Eq. (8). In order to have an unambiguous and deterministic definition, in such cases we can choose among admissible optimal

transport maps the one which minimizes Eq. (8). In fact it is even necessary for it to be a metric. However since this case is quite rare, in our implementation we simply took the first one returned by our sorting algorithm. The complexity of $\mathcal{RPW}_2$ is given by $O(p^2 n \log(n))$ which saves a factor $\frac{M}{p}$ as compared to $\mathcal{PW}_2$ and this term is often greater than 10.

From $\mathcal{RPW}_2$, we define a parametric distance $d_{\boldsymbol{\Theta}}^{\mathcal{RPW}_2}$ between two attributed graphs $(\mathcal{G}, \mathbf{X})$ and $(\mathcal{G}', \mathbf{X}')$ :

$$d_{\boldsymbol{\Theta}}^{\mathcal{RPW}_2}(\mathcal{G}, \mathcal{G}') = \mathcal{RPW}_2(\mathcal{D}_{\boldsymbol{\Theta}}(\mathcal{G}, \mathbf{X}), \mathcal{D}_{\boldsymbol{\Theta}}(\mathcal{G}', \mathbf{X}')) \qquad (9)$$

All the metric learning experiments will be conducted using this distance, excepted in an ablative study where we report the use of $\mathcal{SW}_2$ and $\mathcal{PW}_2$. Note that $d_{\boldsymbol{\Theta}}^{\mathcal{RPW}_2}$ is a pseudo-metric, since there are different graphs that can have the same features outputted by the GCN, hence leading to similar representative distribution, therefore the identity of indiscernibles is not verified.

### 4.3 Loss for training distance: the Nearest Class Cloud Metric Learning

The last element to complete our model is to define the loss function $\mathcal{F}$ for Eq. (1). We propose here a new loss function for the purpose of improving the $k$-nearest neighbors method. Actually there are classical losses already efficient for this purpose: one can notably mention Large Margin Nearest Neighbor (LMNN) [15] and Neighbourhood Component Analysis (NCA)[14]. However, the optimization is done using a gradient descent algorithm. Since computing all pairwise distances between graphs at each step of gradient descent would be intractable for large datasets, we have to train our loss in a batch way. In this context, LMNN may be not relevant since this method works locally and a batch is often not representative of the true neighborhood of an element of the dataset. On the contrary NCA loss can be trained in a batch way, as it relies on a probability model which tends to attract elements with the same label with each other, wherever they are. However, preliminary experiments showed only a slight improvement of the k-NN with NCA. Therefore we have constructed a new loss which proposes a different way to ensure the same condition (see Appendix A.2) and which experimentally works better in our setting (see Ablative study, Sec. 5.4). The model is called Nearest Cloud Class Metric Learning (NCCML); the probability of being labeled by $e \in \mathbb{E}$ for a graph $\mathcal{G}$ depends on the distance to the point clouds of a class (hence the name of the method):

$$p_{\boldsymbol{\Theta}}(e|\mathcal{G}) = \frac{\exp\left(-\sum_{\substack{\mathcal{G}_i \in \mathbb{G} \\ \mathcal{E}(\mathcal{G}_i)=e}} d_{\boldsymbol{\Theta}}^{\mathcal{RPW}_2}(\mathcal{G}, \mathcal{G}_i)^2\right)}{\sum_{e' \in \mathbb{E}} \exp\left(-\sum_{\substack{\mathcal{G}_i \in \mathbb{G} \\ \mathcal{E}(\mathcal{G}_i)=e'}} d_{\boldsymbol{\Theta}}^{\mathcal{RPW}_2}(\mathcal{G}, \mathcal{G}_i)^2\right)}. \qquad (10)$$

Given this probability, we want to construct the distance $d_{\boldsymbol{\Theta}}^{\mathcal{RPW}_2}$ maximizing the probability that the labeled graphs in the dataset have the correct labels, which leads to solve the following problem:

$$\max_{\boldsymbol{\Theta}} \mathcal{F}_{\boldsymbol{\Theta}}^{\mathbb{G}} = \max_{\boldsymbol{\Theta}} \sum_{\mathcal{G}_i \in \mathbb{G}, \mathcal{E}(\mathcal{G}_i) \neq \emptyset} \log p_{\boldsymbol{\Theta}}(\mathcal{E}(\mathcal{G}_i)|\mathcal{G}_i). \qquad (11)$$

By maximizing this loss, we construct a distance which, for each element, favors its relative distance to elements of the same labels compared to those of different labels. This should favor k-NN, especially when $k > 1$. We will show in the experiments that, in this specific context, NCCML exhibits better performance than NCA. More details on NCCML can be found in Appendix A.2.

### 4.4 Computational aspects

We test, in the next Section, the proposed metric learning method with $\mathcal{RPW}_2$ (and $\mathcal{SW}_2$ or $\mathcal{PW}_2$ in ablative studies).

**Optimization.** In terms of optimization, we can differentiate directly with respect to one dimensional distribution parameters of Wasserstein distance, thus we can also differentiate through $\mathcal{RPW}_2$ (Eq. (8)) (and also approximation of $\mathcal{SW}_2$ (Eq. (4)) or $\mathcal{PW}_2$). Self-differentiation techniques can be used on these expressions (see [26]). We implemented our algorithm in `tensorflow`[2]. The

---

[2]The implementation can be found in the supplementary material.

---

**Algorithm 1** SGML: High-level algorithm to build $d_{\Theta^*}^{\mathcal{RPW}_2}$.

---

**Require:** A dataset of attributed graphs $\mathbb{G}$ and their labeling function $\mathcal{E}$.
  **for** each epoch $e \in \{1, \dots, E\}$ **do**
    Build a partition: $\cup_k B_k = \mathbb{G}$ such that $B_k \cap B_{k'} = \emptyset$.
    **for** each batch $B_k$ **do**
      **for** each graph pair $(\mathcal{G}, \mathcal{G}') \in B_k \times B_k$ **do**
        Compute distance $d_{\Theta}^{\mathcal{RPW}_2}(\mathcal{G}, \mathcal{G}')$ (Eq. (9))
      Compute $-\mathcal{F}_{\Theta}^{B_k}$ (Eq. (11)) and apply an iteration of Adam descent algorithm.
  **return** all pairwise distance $d_{\Theta^*}^{\mathcal{RPW}_2}$ in $\mathbb{G}$.

---

minimization of the loss is performed by *batch* and stochastic gradient descent (in particular with the optimizer *Adam* [33]).

**Parameters.** The following default parameters are used (unless otherwise indicated in the text): learning rate $l_r = 0.999 * 10^{-2}$, number of epochs $E = 10$, batch size $B = 8$, and the GCN output features size $p = \min(5, q)$. For experiments involving $\mathcal{SW}_2$ and $\mathcal{PW}_2$, the sampling number is set to $M = 50$ which is a common value used in the literature.

**Time complexity.** Theoretically, the training time is negligible compared to the computation of all pairwise distances; therefore we focus on this last step for the time complexity analysis (see Appendix A.5 for runtimes per dataset). If we denote $\tilde{n}$ the number of average nodes of a graph, the total complexity of this computation with $\mathcal{RPW}_2$ (resp. $\mathcal{SW}_2$) is given by $O(|\mathbb{G}|\tilde{n}(p^2 + \tilde{n}rp) + |\mathbb{G}|^2 p^2 \tilde{n} \log \tilde{n})$ (resp. $O(|\mathbb{G}|\tilde{n}(p^2 + \tilde{n}rp) + |\mathbb{G}|^2 pM\tilde{n} \log \tilde{n})$). The first terms occur for application of GCN and the latest for computing distances. In practice, for not too large $\tilde{n}$ values, a quadratic implementation exploiting vectorization can be faster (see section 5.2). Furthermore, one can see that the GCN becomes the limiting element for scaling (on graph sizes); in practice, the sparsity of the adjacency matrix and the optimizations on GPUs limit this problem. However, it is still an active research topic to determine the less expensive ways to characterize the nodes [34, 35].

**Spatial Complexity.** Our quadratic implementation mentioned above requires to store in memory a tensor of size $O(\tilde{n}^2 p)$ for $\mathcal{RPW}_2$ and $O(\tilde{n}^2 M)$ for $\mathcal{SW}_2$ or $\mathcal{PW}_2$. The sequential implementation has a $O(\tilde{n})$ spatial complexity (more details on these implementations are in Appendix A.3). Anyway for both implementations, for the dataset of graphs considered, SGML is very cheap in terms of memory consumption in regards of actual GPU capability.

## 5 Experiments

### 5.1 Datasets

For the experiments, we use a large panel of data sets from the literature [2][3]: ENZYMES, PROTEINS, IMDB-B, IMDB-M, MUTAG, BZER, COX2 and NCI1. More information on these datasets can be found in Appendix A.4. Additional details about the following experiments can be found in Appendix A.7 for reproducibility. When a dataset has discrete features, they are one-hot encoded.

### 5.2 $\mathcal{RPW}_2$ Running times

We have generated uniform random (normal) distributions with support in $\mathbb{R}^5$ of size ranging from $10^1$ to $10^6$. These sizes of the distributions correspond to graph sizes $n$ (number of nodes). The choice of $\mathbb{R}^5$ is motivated by the usual good performance of ML when performed in small dimension. We compare the running time to compute the distance between these distributions with $\mathcal{W}_2$, $\mathcal{W}_2^e$, ($\mathcal{W}_2$ with entropic regularization parameter $\gamma = 100$), $\mathcal{SW}_2$ using POT [36] library, $\mathcal{PW}_2$ and $\mathcal{RPW}_2$. For $\mathcal{RPW}_2$ we compare both the quadratic and the sequential (`numpy`) implementations we developed. The results can be found on Figure 1. Additional details and results are given in Appendix A.6.

As expected $\mathcal{SW}_2$, $\mathcal{PW}_2$ and $\mathcal{RPW}_2$ are the methods scaling the best: we obtain the expected (quasi) linear slope for all three methods $O(n \log n)$. As soon as $n > 10^4$, These three methods

---

[3]http://graphkernels.cs.tu-dortmund.de

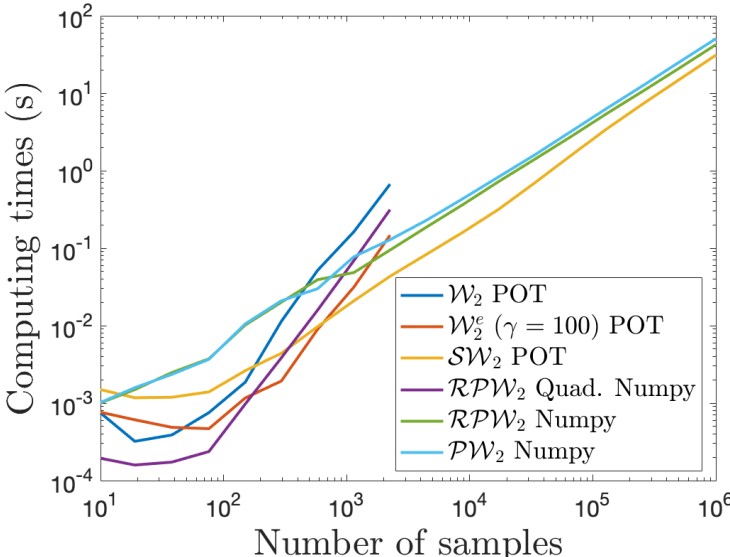

**Figure 1: Run time comparisons.**

allows to compute distances between distributions of several orders of magnitude larger for the same time as $\mathcal{W}_2$ and $\mathcal{W}_2^e$. Although $\mathcal{SW}_2$, $\mathcal{PW}_2$ and $\mathcal{RPW}_2$ scale mostly the same, $\mathcal{SW}_2$ seems a bit faster than $\mathcal{PW}_2$ and $\mathcal{RPW}_2$. However, the slope of $\mathcal{RPW}_2$ is a little bit smaller than $\mathcal{SW}_2$ one. (See Appendix A.6.) Anyway, we will show in the next experiment (Sec. 5.4) that $\mathcal{RPW}_2$ builds better metrics than $\mathcal{SW}_2$. Finally, note that the quadratic implementation of $\mathcal{RPW}_2$ is the fastest for samples with less than 200 instances, which is the case for the datasets considered in the following experiments.

**Table 1: Results of the main experiments for datasets of graphs with discrete attributes.** Features are node labels for NCI1, PROTEINS and ENZYMES; and degrees for others. Accuracy is in bold green when it is the best of its block. For $\mathcal{FGW}$-WL (resp. PSCN), depth is set to 4 (resp. 10).

| Method | MUTAG | NCI1 | PROTEINS | ENZYMES | IMDB-M | IMDB-B |
|---|---|---|---|---|---|---|
| **k-NN** | | | | | | |
| $\mathcal{RPW}_2$ | **90.00 ± 7.60** | 72.12 ± 1.65 | **70.18 ± 4.01** | 49.00 ± 8.17 | 45.00 ± 5.46 | 68.90 ± 5.45 |
| Net-LSD-h | 84.90 | 65.89 | 64.89 | 31.99 | 40.51 | 68.04 |
| FGSD | 86.47 | 75.77 | 65.30 | 41.58 | 41.14 | 69.54 |
| NetSimile | 84.09 | 66.56 | 62.45 | 33.23 | 40.97 | 69.20 |
| **SVM & GCN** | | | | | | |
| $\mathcal{RPW}_2$ | **88.95 ± 7.61** | 74.84 ± 1.81 | 74.55 ± 4.19 | 54.00 ± 7.07 | **51.00 ± 5.44** | **72.00 ± 3.16** |
| WWL | 87.27 ± 1.50 | 85.75 ± 0.25 | 74.28 ± 0.56 | **59.13 ± 0.80** | ✗ | ✗ |
| $\mathcal{FGW}$ | 83.26 ± 10.30 | 72.82 ± 1.46 | ✗ | ✗ | 48.00 ± 3.22 | 63.80 ± 3.49 |
| $\mathcal{FGW}$-WL | 88.42 ± 5.67 | **86.42 ± 1.63** | ✗ | ✗ | ✗ | ✗ |
| WL-OA | 87.15 ± 1.82 | 86.08 ± 0.27 | **76.37 ± 0.30** | 58.97 ± 0.82 | ✗ | ✗ |
| PSCN | 83.47 ± 10.26 | 70.65 ± 2.58 | 58.34 ± 7.71 | ✗ | ✗ | ✗ |

### 5.3 Supervised classification

We evaluate the method in two ways: by using k-NN directly on the computed distances, and by using a SVM with a custom kernel built from the model proposed. We eventually compare the method to several (pseudo-) metric and distances from literature such as NetLSD [37], WWL [12], $\mathcal{FGW}$ [27].

**k-Nearest Neighbors.** Datasets are split in a training (90%) and test set (10%). For each of them we train $\mathcal{RPW}_2$ following Algorithm 1 (in the appendix) on the training set with only one

**Table 2: Results of the main experiments for datasets of graphs with continuous attributes graphs datasets.** The best accuracy are in bold green. Note that for PROTEINS, ENZYMES and CUNEIFORM we concatenate continuous attributes with discrete attributes to build an extended continuous attributes (see Appendix A.7 for more details).

| Method | BZR | COX2 | PROTEINS | ENZYMES | CUNEIFORM |
|---|---|---|---|---|---|
| $\mathcal{RPW}_2$ (kNN) | $\mathbf{85.61 \pm 2.98}$ | $\mathbf{79.79 \pm 2.18}$ | $71.79 \pm 4.47$ | $51.66 \pm 5.16$ | $54.81 \pm 12.26$ |
| **SVM & GCN** | | | | | |
| $\mathcal{RPW}_2$ | $84.39 \pm 3.81$ | $\mathbf{78.51 \pm 0.01}$ | $74.29 \pm 4.11$ | $48.83 \pm 4.78$ | $64.44 \pm 10.50$ |
| WWL | $84.42 \pm 2.03$ | $78.29 \pm 0.47$ | $\mathbf{77.91 \pm 0.80}$ | $\mathbf{73.25 \pm 0.87}$ | ✗ |
| $\mathcal{FGW}$ | $\mathbf{85.12 \pm 4.15}$ | $77.23 \pm 4.86$ | $74.55 \pm 2.74$ | $71.00 \pm 6.76$ | $76.67 \pm 7.04$ |
| PROPAK | $79.51 \pm 5.02$ | $77.66 \pm 3.95$ | $61.34 \pm 4.38$ | $71.67 \pm 5.63$ | $12.59 \pm 6.67$ |
| HGK-SP | $76.42 \pm 0.72$ | $72.57 \pm 1.18$ | $75.78 \pm 0.17$ | $66.36 \pm 0.37$ | ✗ |
| PSCN [K = 10] (GCN) | $80.00 \pm 4.47$ | $71.70 \pm 3.57$ | $67.95 \pm 11.28$ | $26.67 \pm 4.77$ | $25.19 \pm 7.73$ |

hyperparameter to adjust: the depth of SGCN taken as $r = \{1, 2, 3, 4\}$ for all datasets, except for MUTAG for which we go up to 7. The training is done for each parameter $r$ during 10 epochs. A 5-fold cross validation of the number of neighbors $k = \{1, 2, 3, 5, 7\}$ to be considered is performed on the training set using the considered distance. Then for the best $k^*$, we keep the associated validation accuracy, and we finally train a k-NN on the whole training set and evaluate its accuracy on test set. This experiment is averaged on 10 runs. The final test accuracy retained is the one associated to the largest validation accuracy. In this procedure, test set labels were never seen during neither training nor validation. Results are given in the first lines of Table 1 for graphs having labeled nodes and of Table 2 for graphs with continuous attributes.

The learning metric framework combined with k-NN allows us to obtain good performance in classification tasks, in particular for datasets of graphs with continuous attributes. The exception is ENZYMES where we can see a lower net performance. For discrete attributes, SGML performs slightly below the state-of-the-art, yet it outperforms the existing distances classically combined with k-NN. Experiments show that our graph ML distance framework is efficient.

*Note: This procedure is very similar to the one used by WWL, except that the parameter $k$ is replaced by the corresponding parameters of their kernel (see next section).*

**SVM.** To compare to graph kernel methods, the experiment described in the previous section is reproduced using a SVM for classification. The kernel $\boldsymbol{K}_{\mathcal{RPW}_2} = \exp(-\lambda d_{\Theta^*}^{\mathcal{RPW}_2})$ is built from the constructed distance. In this experiment, kernel hyperparameter $\lambda$ and SVM hyperparameter $C$ are tuned similarly as the parameter $k$ above. The set of possible $\lambda$ (resp. $C$) values are 6 (resp. 12) regularly spaced values between $10^{-4}$ and $10^1$ (resp. $10^{-4}$ and $10^5$ including 1). The results are provided in Table 1 (bottom part).

In this part of the table, one can see that the distance learned with our model performs as well as other OT distances when used as a kernel, on the majority of the datasets. We reach or are slightly above state of the art results on 5 datasets over 6 but are still below on NCI1. We recall that our method is specifically designed for the k-nearest neighbors method and that its computational complexity is much lower than many of the best methods on these datasets (notably WWL and $\mathcal{FGW}$).

## 5.4 Ablative study

We perform experiments to justify the design choice of our model. Specifically we show that these choices effectively help to improve k-NN performance by reproducing the experiments above (with k-NN) on different versions of the method without some (or all) of our propositions.

**Raw model.** Without any of our novel propositions, the method would be equivalent to WWL, which corresponds to use the Wasserstein distance between distributions of Eq. (7), where $\boldsymbol{Y}$ is generated with GIN [5], a non trainable GCN. This specific case corresponds to the first column denoted **WWL** of Table 3. We see that even if there are datasets where there is a loss of performance, others benefit from the learned metrics. Moreover we remind that our distance is much less expensive to use than $\mathcal{W}_2$ on which WWL is based.

**Table 3: Ablative study results.** Acc. is the accuracy. $\Delta$ is the difference in accuracy between the model of the column and the proposed one SGML whose results are on Table. 1. Red negative (resp. Green positive) number means that our model performs better (resp. worse).

| Dataset | WWL | | SGML - $\mathcal{SW}_2$ | | SGML - NCA | | SGML - $\mathcal{PW}_2$ | |
|---|---|---|---|---|---|---|---|---|
| Method | Acc. | $\Delta$ | Acc. | $\Delta$ | Acc. | $\Delta$ | Acc. | $\Delta$ |
| **BZR** | 78.05 | - 7.56 | 82.93 | - 2.68 | 83.41 | - 2.20 | 84.39 | - 1.22 |
| **COX2** | 78.51 | -1.26 | 78.30 | - 1.49 | 77.66 | - 2.13 | 78.94 | - 0.85 |
| **MUTAG** | 83.68 | - 6.32 | 86.84 | - 3.16 | 87.37 | - 2.63 | 90.00 | 0.00 |
| **NCI1** | 80.43 | 5.31 | 69.03 | - 3.09 | 69.66 | - 2.46 | 72.90 | 0.78 |
| **PROTEINS** | 71.60 | 1.42 | 71.34 | 1.16 | 71.70 | 1.52 | 70.54 | 0.36 |
| **IMDB-B** | 68.20 | - 0.7 | 68.20 | -0.70 | 67.40 | -1.5 | 68.80 | - 0.10 |
| **IMDB-M** | 48.73 | 3.73 | 42.33 | -2.67 | 42.73 | -2.27 | 44.13 | - 0.87 |
| **ENZYMES** | 56.00 | 7.00 | 44.33 | - 4.67 | 55.33 | 6.33 | 44.83 | -4.17 |

**SGML with $\mathcal{SW}_2$.** This second ablative study is in the second column, denoted **SGML-$\mathcal{SW}_2$**, of Table 3, and is related to replacing $\mathcal{RPW}_2$ by $\mathcal{SW}_2$. The result clearly validates our choice to use $\mathcal{RPW}_2$ instead of $\mathcal{SW}_2$. Our model is the best one except on one dataset.

**SGML with $NCA$.** For this experiment we replaced the loss NCCML by the NCA loss. The result is in the third column, **SGML** - NCA of Table 3. It appears that NCCML is often more appropriate than NCA in our specific ML framework.

**SGML with $\mathcal{PW}_2$.** For this final experiment we used $\mathcal{PW}_2$ instead of $\mathcal{RPW}_2$. This experiments show that $\mathcal{PW}_2$ and $\mathcal{RPW}_2$ have equivalent results. This suggests that projecting only on the canonical basis is sufficiently informative while still being less costly.

Globally, the ablative study is in favor of the choices proposed for SGML. The driving idea in the present article of choosing simple and scalable methods over more complex ones, leads to competitive performance while allowing full scalability.

## 6 Conclusion

In this article, we proposed a metric learning method for attributed graphs, specifically to increase the performance of k-NN. We have shown experimentally that it can indeed achieve performance similar or even superior to the state of the art. However, a theoretical work on the properties of $\mathcal{RPW}_2$ will be useful to allow us to better understand when it does not perform well. Appendix A.9 presents some additional elements on the limits of the work. In addition, further work may easily adapt **SGML** to perform other tasks like graph clustering or regression, with an appropriate (and probably different) ML loss.

## Acknowledgements

The work has been was supported by the ACADEMICS Grant given by the IDEXLYON project of the Université de Lyon, as part of the "Programme Investissements d'Avenir" ANR-16-IDEX-0005, the ANR-19-CE48-0002 DARLING grant, and the GraphNEx CHIST-ERA ANR-21-CHR4-0009 grant.

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
