# OpenReview forum: "A simple way to learn metrics between attributed graphs"
_logconference.io/LOG/2022/Conference — LoG 2022 Poster_

### Official Review · Reviewer_Ymj4 · 2022-10-14

**Overall Score:** 6
**Confidence:** 4

**Review:**

This paper proposes a graph metric learning method with OT. The paper designs a restricted projected slice-Wasserstein distance, and plugs it into a contrastive loss, leading to SGML. The paper also shows some experimental results to show its effectiveness and efficiency.

Strengths:
1. The idea and technique are interesting and make sense.
2. The proposed method is efficient as claimed by the authors.

Weaknesses and questions:
1. RPW2 seems the main contribution of this paper. Therefore, the motivation of RPW2 (i.e., Eq.(8)) should be introduced in more detail. Moreover, the difference between Eq.(8) and Eq.(5) should also be clarified. The only difference is that $n \neq n'$, isn't it?
2. In Table 1, the SVM & GCN results on MUTAG seem strange. Why are the standard deviation (std) so large? Too large std means the results are not stable. From the given results with so large std, although the proposed one achieves the highest mean value (i.e., 88.95), we can hardly say that it outperforms others statistically significantly.
3. From Tables 1 and 2, we can find that the results of the proposed one with k-nn are good, but those with SVM &GCN are not good enough. What are the reasons? Does it mean the proposed one is only appropriate for k-nn?
4. The representation of this paper is not satisfactory enough. There are many typos and syntax errors, e.g. Line 21, "GCN have" -> "GCN has"; Line 29, "on all learning task"  -> "on all learning tasks"; Line 91, "than" -> "that"; Caption of Fig.1, "times" -> "time". The paper needs more careful proofreading.

---

### Official Review · Reviewer_JZFe · 2022-10-18

**Overall Score:** 8
**Confidence:** 3

**Review:**

**Summary**
---

The paper looks at a new definition for computing the distances between graphs that uses both graph neural networks as well as optimal transport. Specifically, the first use a simple GCN (i.e. without intermediate layer non-linearities) to extract embeddings for the attributed graphs. Following this, the paper looks at the uniform distribution on the embeddings and then computes a restricted version of the Projected Wasserstein distance. The restriction here is that we only allowed to do projections along a fixed basis. Once we have this metric, the paper then trains the features in the GCN to optimize the embeddings (and hence the distances) using a Metric Learning loss. Specifically, we place a Gaussian distribution on each graph (using the above defined metric to control the variance) and want to maximize the MLE objective of the labelled points using the labels of the points nearest to it.

The paper then tests the method to do classification and perform an ablation study to determine the effect of the type of optimal transport loss computed and the the use of their loss function.

**Contributions**
---

I think the main contribution of the paper is the complete pipeline built. Each individual piece:

1) Using a GCN to learn features.
2) Treating features as distributions
3) Using OT to learn distance between distributions
4) Using this distance as the parameterized distance that is optimized over using K-NN MLE loss (with Gaussian Kernels)

is not very novel or interesting. However, the manner in which the pieces fit together is novel and interesting. The fact that such a metric turns out to be useful is another plus.

**Strengths**
---

As mentioned in the contributions, I think the major strength of the paper is how the various pieces fit together to give a trainable framework to learn this metric.

The method is fast and provides good empirical evidence that the metric learned is useful.

**Weaknesses**
---

These are mostly minor

---

While I think the ablation study is interesting I think the choice of models compared against is not the best. Specifically, the WWL changes everything (the embedding model, the OT distances and (I think) the loss function as well). The SGML $\mathcal{SW}$ only changes the OT and SGML NCA only changes the loss function. I think the better thing to do would be the following

**Changing Exactly One Piece**

1) SGML - NCA is good as it explored different loss functions.
2) For SGML - $\mathcal{SW}$, I would also like a comparison against SGML - $\mathcal{PW}$. As there two steps of changes between the sliced Wasserstein and the restricted projective Wasserstein.
3) To then complete this part of the ablation study, I would then recommend changing the embedding model. (instead of looking at WWL which changes everything)

Or

**Changing Multiple Pieces**

If we start changing multiple pieces at once, then I think to have a complete ablation study, we would need to look at all 8 subsets of pieces changed. I realize this is a lot of extra work. But having this would provide strong evidence for the choices made.

---

While the presentation of the paper is clear, I think the writing (sentence by sentence) can be improved.

**Questions**
---

1) I did not understand lines 205 to lines 209. Why is this formula not deterministic. Looking at [31] the $pi^{u_k}$ is determined by the coupling given by a certain projection. That is if $x_i$ and $y_i$ are the points, and $v$ is the direction, let $\hat{x}_i = x_i^Tv$ and $\hat{y}_i = y_i^Tv$. Then we sort the $\hat{x}_i$ and $\hat{y}_i$ and match them up. This gives us the coupling. Now I understand two $\hat{x}_i$ may be equal (this seems unlikely numerically) but if they are then we could have different coupling, but then do we not want to pick the one with the smallest distance?

---

### Official Review · Reviewer_vAsD · 2022-10-22

**Overall Score:** 6
**Confidence:** 4

**Review:**

The paper proposes a graph metric learning model based on simple graph convolutional neural networks to learn the hidden representation and apply the 1D optimal transportation from Wasserstein distance. Furthermore, it also proposes a revised softmax loss function over the class point clouds for metric learning.

__Pros__:
* The proposed SGML method claims its scalability due to its quasi-linear complexity on the optimal transport learning with distributions, namely, RPW, which is a revised version of PW.
* The NCCML is a good extension of NCA, especially considering the dataset of graphs (geometric measures).
* The paper is well organized, well written, and has extensive experience.

__Cons(not necessarily the weakness, but some questions to authors)__:
* As joint learning based on both SGCN and the OT, the complexity is not purely based on the OT. Taking the SGCN into the consideration, it’s no longer the quasi-linear complexity in total. Is that correct?
* Again, compare with the SGW [2], a sliced Gromov-Wasserstein also has the quasi-linear complexity $O(nlog(n))$, what’s the advantage of taking the additional SGCN training on graphs? For the experiments, it’s better to include SGW as one of the baselines. The POT package lacks the SGW implementation, but the source code is available to reach.
* The “restricted” PW is taking the projection alongside the base vector of $\mathbb{R}^p$, it’s better to analyze such kind of restriction won’t break the metric measure on 1-D distributions. I don’t have a concrete idea yet, but I doubt missing bases will break the symmetric measures on distributions. More specifically, deriving from eq. (5) to eq. (8) is nothing but a sample with dimension restrictions.
* The specific loss function is designed for the KNN algorithm, will it be possible to expand to other models? For example, with graph kernels (e.g., WL kernel)?

__Further questions (open-ended)__:
* Unlike the FGW or GW has a (quasi)-metric measure on graphs, the proposed work didn’t satisfy the definitions of metrics, as mentioned in the paper, symmetry, identity, and triangle inequality. This is due to the unreliable training on SGCN. Can you illustrate more on this? Or provide some theoretical analysis on these three properties?
* In a high-level view, Wasserstein distance handles the similarities in distributions, while the GW [1] (ignoring the attributed graphs, only structures) /FGW takes the directly optimal transport measure on graphs, which can be considered as a 2D coupling problem, that’s why most of the algorithms in cubic costs. So, no the only advantage of taking SGCN is to transfer the 2D graphs data into 1D distributions, is that correct? Any other approaches?


[1] Peyre et al, “Gromov-Wasserstein Averaging of Kernel and Distance Matrices”. NeurIPS’16

[2] Vayer et al, “Sliced Gromov-Wasserstein”, NeurIPS’19

---

### Meta-Review · Area_Chair_uTEc · 2022-11-15

**Confidence:** 5
**Recommendation:** Accept

**Meta Review:**

This work proposes a metric learning method to learn encoders that measure similarity between input objects. The proposed SGML is simple while preserving good properties such as optimal transport. Reviewers agree that the ideas are novel, and experiments are convincing. Concerns from the reviewers have been adequately addressed by the authors, while there still exist open-ended questions that could potentially lead to future works. I would like to recommend acceptance.

---

### Decision · Program_Chairs · 2022-11-23

Accept (Poster)